# Important Food Sources of Fructose-Containing Sugars and Non-Alcoholic Fatty Liver Disease: A Systematic Review and Meta-Analysis of Controlled Trials

**DOI:** 10.3390/nu14142846

**Published:** 2022-07-12

**Authors:** Danielle Lee, Laura Chiavaroli, Sabrina Ayoub-Charette, Tauseef A. Khan, Andreea Zurbau, Fei Au-Yeung, Annette Cheung, Qi Liu, Xinye Qi, Amna Ahmed, Vivian L. Choo, Sonia Blanco Mejia, Vasanti S. Malik, Ahmed El-Sohemy, Russell J. de Souza, Thomas M. S. Wolever, Lawrence A. Leiter, Cyril W. C. Kendall, David J. A. Jenkins, John L. Sievenpiper

**Affiliations:** 1Department of Nutritional Sciences, Temerty Faculty of Medicine, University of Toronto, Toronto, ON M5S 1A8, Canada; danie.lee@mail.utoronto.ca (D.L.); laura.chiavaroli@alumni.utoronto.ca (L.C.); sabrina.ayoubcharette@mail.utoronto.ca (S.A.-C.); tauseef.khan@utoronto.ca (T.A.K.); andreea.zurbau@mail.utoronto.ca (A.Z.); rodney.auyeung@mail.utoronto.ca (F.A.-Y.); annette.cheung@mail.utoronto.ca (A.C.); qiannie.liu@mail.utoronto.ca (Q.L.); katy.qi@mail.utoronto.ca (X.Q.); amnaz.ahmed@mail.utoronto.ca (A.A.); vivian.choo@wchospital.ca (V.L.C.); sonia.blancomejia@mail.utoronto.ca (S.B.M.); vasanti.malik@utoronto.ca (V.S.M.); a.el.sohemy@utoronto.ca (A.E.-S.); desouzrj@mcmaster.ca (R.J.d.S.); thomas.wolever@utoronto.ca (T.M.S.W.); lawrence.leiter@unityhealth.to (L.A.L.); cyril.kendall@utoronto.ca (C.W.C.K.); david.jenkins@utoronto.ca (D.J.A.J.); 2Toronto 3D Knowledge Synthesis and Clinical Trials Unit, Clinical Nutrition and Risk Factor Modification Centre, St. Michael’s Hospital, Toronto, ON M5C 2T2, Canada; 3INQUIS Clinical Research Ltd. (Formerly GI Labs), Toronto, ON M5C 2N8, Canada; 4Department of Family and Community Medicine, University of Toronto, Toronto, ON M5G 1V7, Canada; 5Department of Nutrition, Harvard T.H. Chan School of Public Health, Boston, MA 02115, USA; 6Department of Health Research Methods, Evidence, and Impact, Faculty of Health Sciences, McMaster University, Hamilton, ON L8S 4K1, Canada; 7Population Health Research Institute, Hamilton Health Sciences Corporation, Hamilton, ON L8L 2X2, Canada; 8Department of Medicine, Temerty Faculty of Medicine, University of Toronto, Toronto, ON M5S 1A8, Canada; 9Division of Endocrinology and Metabolism, Department of Medicine, St. Michael’s Hospital, Toronto, ON M5C 2T2, Canada; 10Li Ka Shing Knowledge Institute, St. Michael’s Hospital, Toronto, ON M5B 1T8, Canada; 11College of Pharmacy and Nutrition, University of Saskatchewan, Saskatoon, SK S7N 5E5, Canada

**Keywords:** alanine aminotransferase, aspartate aminotransferase, intrahepatocellular lipid, non-alcoholic fatty liver disease, sugars, sugar-sweetened beverages

## Abstract

Background: Fructose providing excess calories in the form of sugar sweetened beverages (SSBs) increases markers of non-alcoholic fatty liver disease (NAFLD). Whether this effect holds for other important food sources of fructose-containing sugars is unclear. To investigate the role of food source and energy, we conducted a systematic review and meta-analysis of controlled trials of the effect of fructose-containing sugars by food source at different levels of energy control on non-alcoholic fatty liver disease (NAFLD) markers. Methods and Findings: MEDLINE, Embase, and the Cochrane Library were searched through 7 January 2022 for controlled trials ≥7-days. Four trial designs were prespecified: substitution (energy-matched substitution of sugars for other macronutrients); addition (excess energy from sugars added to diets); subtraction (excess energy from sugars subtracted from diets); and ad libitum (energy from sugars freely replaced by other macronutrients). The primary outcome was intrahepatocellular lipid (IHCL). Secondary outcomes were alanine aminotransferase (ALT) and aspartate aminotransferase (AST). Independent reviewers extracted data and assessed risk of bias. The certainty of evidence was assessed using GRADE. We included 51 trials (75 trial comparisons, *n* = 2059) of 10 food sources (sugar-sweetened beverages (SSBs); sweetened dairy alternative; 100% fruit juice; fruit; dried fruit; mixed fruit sources; sweets and desserts; added nutritive sweetener; honey; and mixed sources (with SSBs)) in predominantly healthy mixed weight or overweight/obese younger adults. Total fructose-containing sugars increased IHCL (standardized mean difference = 1.72 [95% CI, 1.08 to 2.36], *p* < 0.001) in addition trials and decreased AST in subtraction trials with no effect on any outcome in substitution or ad libitum trials. There was evidence of influence by food source with SSBs increasing IHCL and ALT in addition trials and mixed sources (with SSBs) decreasing AST in subtraction trials. The certainty of evidence was high for the effect on IHCL and moderate for the effect on ALT for SSBs in addition trials, low for the effect on AST for the removal of energy from mixed sources (with SSBs) in subtraction trials, and generally low to moderate for all other comparisons. Conclusions: Energy control and food source appear to mediate the effect of fructose-containing sugars on NAFLD markers. The evidence provides a good indication that the addition of excess energy from SSBs leads to large increases in liver fat and small important increases in ALT while there is less of an indication that the removal of energy from mixed sources (with SSBs) leads to moderate reductions in AST. Varying uncertainty remains for the lack of effect of other important food sources of fructose-containing sugars at different levels of energy control.

## 1. Introduction

Sugars consumption has been implicated in the development of non-alcoholic fatty liver disease (NAFLD), the leading form of chronic liver disease worldwide [1] and its associated downstream metabolic diseases including obesity, type 2 diabetes, hypertension, and dyslipidemia [2,3]. There is a specific focus on fructose because of its unique metabolism where it bypasses negative feedback control and acts as an unregulated substrate for de novo lipogenesis, which can stimulate liver fat accumulation [4,5]. Clinical Practice guidelines for the prevention and management of NAFLD recommend a reduction in fructose in addition to weight loss and increased exercise [6].

The evidence to support recommendations to reduce fructose derives largely from trials of overfeeding using sugar-sweetened beverages. An earlier systematic review and meta-analysis of controlled feeding trials [7] of sugar-sweetened beverages (SSBs) showed that when consumed at high doses providing excess calories, there was a significant increase in the NAFLD markers intrahepatocellular lipid (IHCL) and alanine aminotransferase (ALT). However, when fructose-containing SSBs were consumed in matched isocaloric substitution for other carbohydrates, there was no effect on NAFLD markers [7].

It is unknown whether the effects seen for SSBs providing excess energy hold for other important food sources of fructose-containing sugars at different levels of energy control on NAFLD markers. There is evidence on other cardiometabolic outcomes that the harmful effects seen for SSBs do not hold for other food sources of fructose-containing sugars, whereas some (fruit) may even show benefit [8,9,10,11].

To investigate the role of food source and energy control in the prevention and management of NAFLD, we conducted a systematic review and meta-analysis of controlled trials to assess the certainty of evidence for the effect of different food sources of fructose-containing sugars at different energy control levels on intrahepatocellular lipid (IHCL), as well as alanine aminotransferase (ALT), and aspartate aminotransferase (AST).

## 2. Materials and Methods

We followed the Cochrane Handbook for Systematic Reviews of Interventions (v6.1) [12] for the conduct of our systematic review and meta-analysis and reported our results following the Preferred Reporting Items for Systematic Reviews and Meta-Analyses (PRISMA) guidelines [13] (Appendix A). The study protocol is registered at ClinicalTrials.gov (NCT02716870).

### 2.1. Data Sources and Search Strategy

We conducted a systematic search in MEDLINE, Embase, and the Cochrane Central Register of Controlled Studies databases through 7 January 2022. Appendix A present the search strategy based on the PICOTS framework, there were no language restrictions. Validated filters from the McMaster University Health Information Research Unit were applied to limit the database search to controlled studies only [14]. The searches were supplemented with manual searches of the reference list of included studies.

### 2.2. Study Selection

We included randomized and non-randomized controlled feeding trials in humans of all health backgrounds and ages, with intervention periods ≥7 days investigating the effect of orally consumed fructose-containing sugars from various food sources compared with control diets that were free of fructose or at least 5 g lower in fructose-containing sugars on at least one NAFLD marker (IHCL, ALT, or aspartate aminotransferase (AST)). We excluded studies of liquid meal replacement interventions and studies of interventions or comparators of rare sugars that contain fructose (e.g., isomaltulose, melezitose, or turanose) or were low calorie epimers of fructose (e.g., allulose, tagatose, sorbose). Reports were initially excluded based on review of their titles and abstracts by a single reviewer. Those reports that remained were then excluded based on review of the full text reports by at least 2 reviewers (DL, LC, SAC, FAY, AC, QL, XQ, AA) independently, leaving the final set of reports to be included in our syntheses. We prespecified four study designs based on energy control: (1) ‘substitution’ trials, in which energy from the food sources of fructose-containing sugars was substituted for other non-fructose-containing macronutrients under energy matched conditions; (2) ‘addition’ trials, in which excess energy from the food sources of fructose-containing sugars was added to the background diet compared to the same diet alone without the excess energy (with or without the use of non-nutritive/low-calorie sweeteners to match sweetness); (3) ‘subtraction’ trials, in which energy from the food sources of fructose-containing sugars was subtracted from background diets compared with the original background diets through displacement by water or low-calorie sweeteners or elimination altogether; and (4) ‘ad libitum’ trials, in which energy from the food sources of fructose-containing sugars was freely replaced (usually within reasonable limits, e.g., intake required to be between 75 and 125% of predicted daily energy requirements) with other non-fructose-containing macronutrients without any strict control of either the study foods or the background diets, allowing for free replacement of energy. In reports containing more than one eligible trial comparison, we included them as separate trial comparisons.

### 2.3. Data Extraction

At least two reviewers (DL, LC, SA-C) independently extracted data from eligible studies. Relevant information included food source of fructose-containing sugars, number of participants, setting, participant health status, study design, level of feeding control, randomization, comparator, fructose-containing sugars type, macronutrient profile of the diets, follow-up duration, energy balance, funding source and outcome data. Appendix A shows the code definitions for the different food sources of fructose-containing sugars. Authors were contacted for missing outcome data when it was indicated that an NAFLD marker was measured but not reported. In the absence of outcome data and inability to obtain the original data from authors, values were extracted from figures using Plot Digitizer [15] where available.

### 2.4. Risk of Bias Assessment

Included studies were assessed for risk of bias independently and in duplicate by at least 2 reviewers (DL, LC, SA-C) using the Cochrane Risk of Bias Tool [12]. Assessment was done across six domains of bias (sequence generation, allocation concealment, blinding, incomplete outcome data, selective outcome reporting and other). Risk of bias for each domain was assessed as either “low” (proper methods taken to reduce bias), “high” (improper methods creating bias) or “unclear” (insufficient information provided). The “other” domain applied only to crossover trials, “high” risk of bias was given when there was no washout between interventions, otherwise the trial was rated as “low”. Reviewer discrepancies were resolved by consensus or arbitration by the senior author (JLS).

### 2.5. Outcomes

The primary outcome was IHCL. Secondary outcomes were ALT and AST. Mean differences (MDs) between the intervention and control arm and their standard errors (SEs) were extracted for each eligible trial comparison. If unavailable, they were derived from available data using published formulas [12]. Mean pairwise difference in change-from-baseline values were preferred over end values. When median data were provided, they were converted to mean data with corresponding variances using methods developed by Luo et al. (2018) [16] and Wan et al. (2014) [17]. When no variance data were available, the standard deviation (SD) was borrowed from a trial similar in size, participants and nature of intervention [18].

### 2.6. Data Syntheses and Analyses

We used STATA software, v16.1 (StataCorp, College Station, TX, USA) for all analyses. As our primary research question was to assess the effect of different food sources of fructose-containing sugars at different energy control levels, we performed separate pairwise meta-analyses for each of the four prespecified designs by energy control level (substitution, addition, subtraction and ad libitum trials) and assessed the interaction between food sources of fructose-containing sugars within each energy control level using the Cochrane Handbook’s recommended standard Q-test for subgroup differences (significance at *p* < 0.10) [19,20,21].

The principal effect measures were the mean pair-wise differences in change from baseline (or alternatively, end differences) between the food sources of fructose-containing sugars arm and the comparator arm (significance at *p* < 0.05). Data were analyzed using the generic inverse variance method with DerSimonian and Laird random-effects model [12,22]. A fixed effects model was used when only <5 trial comparisons were available [17]. Paired analyses were applied to all crossover trials with the use of a within-individual correlation coefficient between treatment of 0.5 [23,24,25]. Data were expressed as standardized mean differences (SMDs) for IHCL since the available data were reported in various units, and MDs for ALT and AST, with 95% confidence intervals (CIs). Standardized mean difference (SMD) was calculated for each study where the difference between two treatment and control means were divided by the standard deviation pooled across both groups [26]. A pooled SD from baseline reporting % liver fat was used to convert the IHCL SMD to % liver fat for clinical interpretation. To mitigate a unit-of-analysis error, when arms of trials with multiple intervention or control arms were used more than once, the corresponding sample size was divided by the number of times it was used for calculation of the standard error [12].

Heterogeneity was assessed by visual inspection of the forest plots and using the Cochran Q statistic and quantified using the *I*^2^ statistic [12]. We considered an *I*^2^ ≥ 50% and P_Q_ < 0.10 as evidence of substantial heterogeneity [12]. Sources of heterogeneity were explored by sensitivity analyses, including influence analysis, altering pairwise comparison correlation coefficient and subgroup analyses. The influence analysis systematically removed each trial comparison from the meta-analysis with recalculation of the summary effect estimate. A trial whose removal explained the heterogeneity or changed the significance, direction, or magnitude (by more than the minimally important difference (MIDs) for IHCL (5% of the baseline in SMD units (0.26) due to variability in assessment measurement tools), ALT (2.85 U/L) [27], and AST (2.55 U/L) [27] of the effect was considered an influential trial. To determine whether the overall results were robust to the use of different correlation coefficients in crossover trials, we also conducted sensitivity analyses using correlation coefficients of 0.25 and 0.75. If ≥10 trials were available [20,28] we conduced subgroup analyses to explore sources of heterogeneity using meta-regression (significance at *p* < 0.05). A priori subgroup analyses were conducted by participant health status, age, randomization, energy balance, baseline outcome levels, fructose sugars type, comparator type, study design, follow-up, feeding control, fructose-containing sugars dose, and funding. Post-hoc subgroup analyses were conducted by sugars regulatory designation and type of imputation performed for deriving variances. Meta-regression analyses were used to assess the significance of each subgroup categorically and, when applicable, as a continuous exposure.

If ≥6 trial comparisons were available [29], then we assessed linear and non-linear (restricted cubic splines) dose–response relationships (significance at *p* < 0.05) using meta-regression. We also assessed non-linear dose–response threshold effects with three prespecified spline knots at public health thresholds of 5% [30,31], 10% [31,32], and 25% [33] total energy from fructose-containing sugars.

If ≥10 trials were available, then we assessed publication bias by visual inspection of contour-enhanced funnel plots and formal testing with Egger’s [34] and Begg’s [35] tests (significance at *p* < 0.10) [36]. If there was evidence of publication bias, then we adjusted for funnel plot asymmetry and assessed for small-study effects by imputing the missing trial data using the Duval and Tweedie trim-and-fill method [37].

### 2.7. Certainty of the Evidence

The certainty of the evidence was assessed using the Grading of Recommendations, Assessment, Development, and Evaluation (GRADE) approach and software (GRADEpro GDT, McMaster University and Evidence Prime Inc., Hamilton, ON, Canada) [38]. The assessments were conducted by at least two independent reviewers and discrepancies were resolved by consensus or arbitration by the senior author (JLS). The evidence was rated as high, moderate, low, or very low certainty. The included controlled trials were initially rated as high certainty by default and then downgraded or upgraded based on pre-specified criteria. Reasons for downgrading the evidence included risk of bias (assessed by the Cochrane Risk of Bias Tool [12]), inconsistency (substantial unexplained inter-study heterogeneity, *I*^2^ > 50% and *p* < 0.10), indirectness (presence of factors that limit the generalizability of the results), imprecision (the 95% CI for effect estimates overlap the MIDs for benefit or harm), and publication bias (significant evidence of small study effects). The reason for upgrading the evidence was presence of a significant dose–response gradient [39,40,41,42,43,44].

## 3. Results

### 3.1. Search Results

Figure 1 shows the flow of the literature. We retrieved 4616 reports from databases and manual searches, 4433 of which were excluded based on the title or abstract. Of the 174 reports reviewed in full text, 51 reports of controlled feeding trials (75 trial comparisons, *n* = 2059) met the eligibility criteria [45,46,47,48,49,50,51,52,53,54,55,56,57,58,59,60,61,62,63,64,65,66,67,68,69,70,71,72,73,74,75,76,77,78,79,80,81,82,83,84,85,86,87,88,89,90,91,92,93,94]. These trials included ten different food sources of fructose-containing sugars (SSBs; sweetened dairy alternative (soy); 100% fruit juice; fruit; dried fruit; mixed fruit forms; sweets and desserts; added nutritive (caloric) sweetener; honey; and mixed sources (with SSBs)) across four energy control levels: substitution (35 trial comparisons); addition (39 trial comparisons); subtraction (4 trial comparisons); and ad libitum (2 trial comparisons). The mixed sources food category includes those trials in which the intervention included more than one of the food sources (e.g., SSBs and fruits). For the present meta-analysis, all trials categorized as mixed sources included SSBs in the dietary prescription. Out of the ten authors who were contacted for missing NAFLD outcome data, five responded and provided unpublished data [47,49,69,70,81].

### 3.2. Trial Characteristics

Table 1 and Appendix A show the trial characteristics. Trial sizes ranged from a median of 12 participants (range 5–15) in subtraction trials to 92 participants (range 92–92) in ad libitum trials. Participants were predominantly healthy mixed weight or overweight/obese (based on BMI criteria) with some participants who had NAFLD or diabetes. There were proportionally more males than females for subtraction and addition trials, but proportionately more females for subtraction and ad libitum. Most participants were younger adults with ages ranging from a median of 28.7 (range: 28–29) years in subtraction trials to 38 (range: 8–59) years in substitution trials. Most trials were conducted in an outpatient setting (50–100%) and performed in European countries. Most substitution and ad libitum trials were parallel (62% and 100%, respectively), whereas addition trials were mostly crossover (56%) and subtraction trials were equal in crossover and parallel design. Feeding control was mostly supplemented for substitution (66%), addition (87%), subtraction (50%), and all ad libitum (100%) trials. Most studies were randomized (50–91%), whereas ad libitum trials were non-randomized (100%). The dose of fructose-containing sugars ranged from a median of 12% (1–42%) in addition trials to 20% (1–42%) of total energy intake in substitution trials. The follow-up duration ranged from a median of four weeks (range 1–52 weeks) in substitution trials to 88 weeks (88–88) in ad libitum trials. The comparators for substitution trials were mostly glucose (12/35, 34%), diet alone for addition trials (22/39, 56%), and non-nutritive sweetener for all subtraction and ad libitum trials (4/4, 100% and 2/2, 100%, respectively). SSBs was the main food source for substitution (13/35, 37%), addition (20/39, 51%), and subtraction (4/4, 100%) trials, whereas mixed sources were used in all ad libitum trials (2/2, 100%). Most trials were funded by agency sources across for substitution and addition trials (government, not-for-profit health agency, or university sources) (54% and 69%, respectively), agency and industry sources for subtraction trials (50%), and no funding was reported for ad libitum trials.

### 3.3. Risk of Bias

Appendix A show a summary of the risk of bias assessments of the included trials. Most trials were assessed as having unclear risk of bias in random sequence generation (41/75, 55%), allocation concealment (46/75, 61%), and incomplete outcome data domains (47/75, 63%) due to poor reporting, while most were low risk of bias in blinding (47/75, 63%), selective outcome reporting (47/75, 63%), and other (carry-over effects) risk of bias (62/75, 83% domains. Few studies were assessed as having high risk of bias, for random sequence generation (12/75, 12%), allocation concealment (15/75, 20%), blinding of participants and personnel (3/75, 4%), incomplete outcome data (0/75, 0%), selective outcome reporting (2/75, 3%), and other (carry-over effects) (13/75, 17%) risk of bias domains, respectively. Thus, there was no overall serious risk of bias in any trial comparisons except for ad libitum trial comparisons which had 100% (both two trial comparisons) high risk of bias rating for sequence generation and allocation concealment.

### 3.4. Primary Outcome

Figure 2 and Appendix A present the effect of different food sources of fructose-containing sugars on IHCL at four levels of energy control. Total fructose-containing sugars resulted in a large increase in IHCL for addition trials (13 trials; SMD: 1.72; 95% CI: 1.08, 2.36, P_SMD_ < 0.001; no heterogeneity, *I*^2^ = 0.00%, P_Q_ = 0.943), whereas there was no effect in substitution trials (16 trials; SMD: 0.36; 95% CI: −0.07, 0.79; P_SMD_ = 0.098; no substantial heterogeneity, *I*^2^ = 6.70%, P_Q_ = 0.377) or subtraction trials (2 trials; SMD: −0.52; 95% CI: −1.60, 0.56; P_SMD_ = 0.345; no heterogeneity, *I*^2^ = 0.00%, P_Q_ = 0.470). No ad libitum trials were available for IHCL.

There was evidence of influence by food source on IHCL in addition and subtraction trials. Although we were unable to assess interaction by food source at these levels of energy control, the large increase in IHCL in the addition trials and no effect in IHCL in the subtraction trials was specific to a single food source: SSBs. There was no evidence of interaction by food source in the substitution trials (*p* = 0.665).

### 3.5. Secondary Outcomes

Figure 3 and Appendix A present the effect of different food sources of fructose-containing sugars on ALT. There was no effect of total fructose-containing sugars on ALT in substitution trials (28 trials; MD: −0.37 U/L; 95% CI: −1.71, 0.97; P_MD_ = 0.589; substantial heterogeneity, *I*^2^ = 52.64%, P_Q_ = 0.001), addition trials (31 trials; MD: 0.91 U/L; 95% CI: −0.39, 2.21; P_MD_ = 0.169; no substantial heterogeneity, *I*^2^ = 31.44%, P_Q_ = 0.050), subtraction trials (4 trials; MD: −4.86 U/L; 95% CI: −15.91, 6.19; P_MD_ = 0.388; no substantial heterogeneity, *I*^2^ = 38.84%, P_Q_ = 0.179), or ad libitum trials (2 trials; MD: 1.02 U/L; 95% CI: −0.87, 2.92; P_MD_ = 0.290; no heterogeneity, *I*^2^ = 0.00%, P_Q_ = 0.509).

In addition trials, although the test for interaction by food sources was not significant (*p* = 0.159), we considered there to be an influence by food source on ALT as SSBs contributed the majority (42.4%) of the weight in the overall analysis. SSBs (12 trials; MD: 3.09 U/L; 95% CI: 0.49, 5.68 U/L; P_MD_ = 0.020; substantial heterogeneity, *I*^2^ = 58.18%, P_Q_ = 0.006) resulted in an increase in ALT, but other food sources were not significant and showed variable directions of effect estimates. There was evidence of influence by food source on ALT in subtraction trials (*p* = 0.061), although neither food source (SSBs nor mixed sources [with SSBs]) showed an effect on ALT. Even though we were unable to assess an interaction by food source in ad libitum trials, the non-significant effect was specific to a sole food source: mixed sources (with SSBs). There was no evidence of interaction by food source in substitution trials (*p* = 0.584).

Figure 4 and Appendix A present the effect of different food sources of fructose-containing sugars on AST.

Total fructose-containing sugars resulted in a decrease in AST for subtraction trials (4 trials; MD: −5.18 U/L; 95% CI: −8.60, −1.76; P_MD_ = 0.003; no substantial heterogeneity *I*^2^ = 15.10%, P_Q_ = 0.316). There was no significant effect of total fructose-containing sugars independent of food source on AST in substitution (23 trials; MD: 0.39 U/L; 95% CI: −0.87, 1.65; P_MD_ = 0.546; no substantial heterogeneity, *I*^2^ = 46.64%, P_Q_ = 0.008), addition (21 trials; MD: −0.03 U/L; 95% CI: −0.82, 0.76; P_MD_ = 0.945; no substantial heterogeneity *I*^2^ = 7.84%, P_Q_ = 0.357), or ad libitum (2 trials; MD: −0.45 U/L; 95% CI: −1.26, 0.36; P_MD_ = 0.278; no heterogeneity *I*^2^ = 0.00%, P_Q_ = 0.716) trials.

There was evidence of interaction by food source on AST in addition trials (*p* = 0.007), although none of the food sources showed an effect on AST. We considered there to be evidence of influence by food source in subtraction trials was mixed sources (with SSBs) showed significant reduction in AST (2 trials; MD: −7.33 U/L; 95% CI: −12.78, −1.87 U/L; P_MD_ = 0.009; no substantial heterogeneity, *I*^2^ = 31.00%, P_Q_ = 0.229) while SSBs alone did not. In ad libitum trials, the non-significant effect was specific to a sole food source: mixed sources (with SSBs). There was no significant effect of interaction by food source in the substitution trials (*p* = 0.620).

### 3.6. Sensitivity Analyses

Appendix A present the influence analyses for total fructose-containing sugars at the four levels of energy control. The systematic removal of a single trial provided partial explanation of the evidence of substantial heterogeneity for ALT in substitution trials [56,78,81].

Appendix A present the influence analyses for individual food sources for those analyses that showed evidence of an interaction or influence by food source. Removal of a single trial resulted in a change in direction of the effect of ALT in subtraction trials [48] and AST in substitution [68] and addition trials [68].

Appendix A shows sensitivity analyses in which different correlation coefficients (0.25 and 0.75) were used for paired analyses of crossover trials. The use of these different correlation coefficients did not alter the direction, magnitude, or significance of the effect or evidence for heterogeneity for any outcomes across food sources and levels of energy control. The three exceptions where the use of a correlation of 0.75 led to substantial heterogeneity for the effect of mixed sources (with SSBs) on SSBs on AST addition trials (9 trials, *I*^2^ = 51.46%, P_Q_ = 0.036).

Appendix A present the subgroup analyses for all outcomes except subtraction and ad libitum trials as <10 trial comparisons were available for analyses. For the primary outcome IHCL, there was no evidence of significant effect modification by any subgroup category. Secondary outcomes ALT and AST showed significant effect modification in at least one of the following: health status, age, baseline outcome, randomization, energy balance, fructose sugars type, comparator, study design, follow-up, feeding control, dose, regulatory designation, funding, type of mean difference, sugar form matrix, data source, sequence generation, allocation concealment, blinding, incomplete outcome data, selective outcome data, and other carry-over effects. Subgroup analyses by addition ALT trials for SSBs showed no significant effect modification by any subgroup category except for follow-up, feeding control, and blinding. No subgroup analyses were performed for any other individual food sources where there was a significant interaction or influence by food source or when SSBs or mixed sources were the sole food sources as <10 trials were available in these analyses.

### 3.7. Dose–Response Analyses

Appendix A present linear and non-linear dose–response analyses. There was no dose response for the effect of total fructose-containing sugars on IHCL or AST in substitution and addition trials. A significant positive linear dose–response gradient was found for the effect of total food sources of fructose-containing sugars on ALT in addition trials (coef_linear_ = 0.153 U/L (0.035, 0.271) per 1%E, P_linear_ = 0.011). No significant dose response was detected for the effect of any food source which was assessed (≥6 trial comparisons). Dose–response analyses were not performed on subtraction and ad libitum trials (<6 trials).

### 3.8. Publication Bias

Appendix A present the publication bias assessments for all outcomes. There was evidence of funnel plot asymmetry and small study effects for IHCL, ALT and AST in substitution and addition trials (Begg’s test *p* < 0.05; Egger’s test *p* < 0.05). Adjustment for funnel plot asymmetry with the imputation of missing trials by the Duval and Tweedie trim-and-fill method, however, did not alter the direction, magnitude, or significance of the effect, suggesting that there was no meaningful influence of publication bias on the results. No evidence of publication bias was detected in addition ALT trials by SSBs. Publication bias could not be assessed in subtraction or ad libitum comparisons, or for food sources where there was significant interaction by food source, as there were <10 trials available for these analyses.

### 3.9. GRADE Assessment

Figure 2, Figure 3 and Figure 4 and Appendix A present the GRADE assessments. The certainty of evidence for the effect of total fructose-containing sugars on IHCL was moderate in substitution trials, low in addition trials, and very low in subtraction trials, owing to double downgrades for indirectness in the addition and subtraction trials and single downgrades for imprecision in the substitution and subtraction trials. The certainty of evidence for the effect of total fructose-containing sugars on secondary outcomes was high for the lack of effect on ALT and AST in the substitution trials and low to very low for the lack of effect on ALT and AST across the other levels of energy control, with the exception of the reduction on AST in subtraction trials, owing to double downgrades for indirectness and a single downgrade for risk of bias or imprecision.

Because SSBs were the sole food source in addition and subtraction trials for IHCL, and we detected an interaction or influence by food source in the addition, subtraction, and ad libitum trials for ALT and AST, we assessed the certainty of evidence for each of the food sources in these analyses. The certainty of evidence was high for the effect of SSBs (large increase) on IHCL in the absence of any downgrades, and moderate for the effect of SSBs (small important increase) on ALT, owing to a single downgrade for imprecision in addition trials; and low for the effect of the removal of mixed sources (with SSBs) (moderate reduction) on AST owing to downgrades for serious risk of bias and indirectness in subtraction trials. For all other food sources, the certainty of evidence varied from moderate to very low owing to downgrades for risk of bias, indirectness, and/or imprecision.

## 4. Discussion

We conducted a systematic review and meta-analysis of 51 studies (75 trial comparisons) in 2059 predominantly healthy mixed weight or overweight/obese younger adult participants that assessed the effect of 10 different food sources of fructose-containing food sources (sugar-sweetened beverages [SSBs]; sweetened dairy alternative; 100% fruit juice; fruit; dried fruit; mixed fruit sources; sweets and desserts; added nutritive sweetener; honey; and mixed sources [with SSBs]) with a median dose of 12% (range 1% to 35%) to 20% (range 1% to 42%) of total energy intake across four different levels of energy control over a median follow-up of 1 to 88 weeks. We showed that total fructose-containing sugars led to a clinically meaningful, moderate relative increase of 10% from baseline IHCL levels in IHCL in addition trials, whereas there was a moderate reduction of 19% in AST in subtraction trials. Total fructose-containing sugars had no other effects. There was evidence of influenced by food source in these analyses. SSBs providing 27% to 30% excess energy led to a moderate increase in IHCL of 10% (as the sole food source) and a small important increase in ALT of 11% in addition trials, whereas the removal of energy from mixed sources (with SSBs) resulted in a moderate reduction of 28% in AST in subtraction trials.

### 4.1. Findings in Relation to the Literature

Our findings are in agreement with a previous systematic review and meta-analysis by Chiu et al. 2014 [7] in which addition studies of fructose-containing sugars, in the form of SSBs, were found to harmfully affect IHCL and ALT levels. Our results show harm for IHCL which relates to ~0.7% liver fat as well as a significant harmful increase in ALT in SSB addition trials. Chiu et al. [7] also demonstrated that in substitution trials, where energy is matched, fructose-containing sugars from SSBs did not significantly change IHCL and ALT, which corresponds with our findings. Another systematic review and meta-analysis by Chung et al. 2014 [95] found a non-significant difference on ALT and AST in addition trials comparing fructose and glucose diets. We also did not find a significant effect on AST in addition trials of SSBs, and this may be due to the fewer number of studies available (9 vs. 13 and 12 in the IHCL and ALT analyses, respectively).

Our data are also congruent with several systematic review and meta-analyses of prospective cohort studies that investigated the effects of elevated fructose-containing SSB consumption on NAFLD risk. A meta-analysis [96] of seven observational studies (six cross-sectional and one cohort study, *n* = 4639) found that compared with those who did not consume SSB, up to one serving per day was associated with a 53% increased risk of NAFLD (risk ratio 1.53 (95% CI: 1.34, 1.75). Similarly, a systematic review and meta-analysis of four cross-sectional studies (*n* = 6326) found that daily consumption of SSBs was associated with a 40% increased risk in NAFLD compared to the group not consuming any SSBs [97]. In contrast, an observational study in older Finnish adults demonstrated an inverse association between NAFLD risk and fructose intake [98], where the main food source of fructose-containing sugars was consumed in the form of fruit rather than SSBs. We found no significant effect of fruit on any NAFLD markers, though our confidence in these effect estimates is limited due to indirectness and imprecision, and only included data from one to three small trials.

### 4.2. Potential Mechanisms

Appendix A shows several mechanisms that have been proposed to explain the effects of fructose-containing sugars and NAFLD markers. Briefly, the consumption of SSBs as excess calories may stimulate disruptions in hepatic fatty acid metabolism, the various effects shown could be due to the different food matrices/forms of food sources of fructose-containing sugars, and subtraction trials resulting in no effect on NAFLD outcomes could be due to energy compensation.

### 4.3. Strengths and Limitations

Our systematic review and meta-analysis has several strengths. First, the literature search and selection methods were comprehensive and reproducible. Second, the synthesized data were derived from all the available evidence from controlled interventions, providing a large body of evidence (51 studies, 75 trials comparisons, *n* = 2059), which minimizes bias and increases precision. Third, our approach included a comprehensive exploration of possible sources of heterogeneity. Fourth, none of our analysis had substantial unexplained heterogeneity. Fifth, we evaluated the shape and strength of the dose–response relationships. Finally, the overall certainty of the evidence was assessed by GRADE.

There were also several limitations. First, there was evidence of serious risk of bias in the ad libitum trials of ALT and AST. The two available ad libitum trials were neither randomized nor concealed the allocation of treatment groups to participants, thereby increasing the risk of bias in the measured effects. Second, there was evidence of very serious indirectness in several analyses. There was significant interaction or influence by food source for ALT and AST in addition trials and thus we double downgraded total fructose-containing sugars in these analyses for indirectness. Another source of indirectness was the limited number of food sources of fructose-containing sugars available for some analyses. SSBs and/or mixed sources (with SSBs) was the sole food source in addition and subtraction trials of IHCL and ad libitum trials for ALT and AST. The lack of alternate food sources limits our ability to explore interactions by food source and is thus unclear whether these effects hold for other important fructose-containing food sources. In some analyses with few trials, we downgraded for indirectness due to limited generalizability. Another source of indirectness may have been that we were unable to differentiate between NAFLD and non-alcoholic steatohepatitis (NASH) as the effects of fructose-containing sugars could differ based on NAFLD status. All obtained studies with NAFLD participants did not diagnose for NASH except Vos et al. 2009 [83], where seven pediatric participants were confirmed to have NASH from a liver biopsy and there was no effect between the NASH and NAFLD participants. Finally, there was evidence of serious imprecision in the effect of total fructose-containing sugars on IHCL in the substitution and subtraction trials with the 95% confidence intervals crossing the prespecified minimally important differences on either side of unity. Thus, we cannot rule out clinically important benefit and/or harm for these NAFLD outcomes.

Weighing the strengths and limitations, the certainty of the evidence was high and moderate for the increasing effect of SSBs providing excess energy on IHCL and ALT, respectively, in addition trials, low for the decreasing effect of the removal of energy from mixed sources (with SSBs) on AST, and generally low to moderate for all other comparisons.

### 4.4. Implications

Our findings have implications for public health and clinical practice. General dietary guidelines and clinical practice guidelines for obesity, diabetes, and cardiovascular disease have shifted away from a focus on single nutrients to a focus on foods and dietary patterns [99,100,101,102]. Clinical practice guidelines for NAFLD have also shifted in this direction recommending a Mediterranean dietary pattern, although there is still a recommendation to avoid fructose intake irrespective of the food source [6]. Understanding the differential effects of food sources of fructose-containing sugars on NALFD markers aligns with this approach to nutrition and health in the prevention and management of NAFLD. The evidence supports reducing the consumption of SSBs as a source of excess energy, while it does not support avoidance of food sources such as 100% fruit juice, dried fruit, mixed fruit, or plant-based dairy alternative, all of which are part of one or more therapeutic dietary pattens such as Mediterranean, Dietary Approaches to Stop Hypertension (DASH), vegetarian, Portfolio, and low-glycemic index dietary patterns [102].

## 5. Conclusions

In conclusion, the effect of fructose-containing sugars on markers of NAFLD appears mediated by both energy control and food source in predominantly mixed weight or overweight/obese younger adults. The evidence provides a good indication that SSBs providing excess energy at high doses lead to large increases liver fat and small important increases in ALT, whereas there is a less of an indication that the removal of energy from mixed sources (with SSBs) results in moderate reductions in AST. Varying uncertainty remains for the lack of effect of other important food sources of fructose-containing sugars such as fruit, 100% fruit juice, dried fruit, mixed fruit sources, sweetened dairy alternatives, sweets and desserts, and mixed sources at different levels of energy control. The main sources of uncertainty in the evidence are imprecision and indirectness (most of the interventions were limited to SSBs). There is a need for more large high-quality randomized trials that assess a broader variety of food sources of fructose-containing sugars (e.g., fruit, dairy and grain-based products) should be explored in future studies at different levels of energy control to address these issues. In the meantime, our findings suggest policy and guideline makers should consider the role of energy and food matrix in recommendations for sugars in NAFLD prevention and management.

## Figures and Tables

**Figure 1 nutrients-14-02846-f001:**
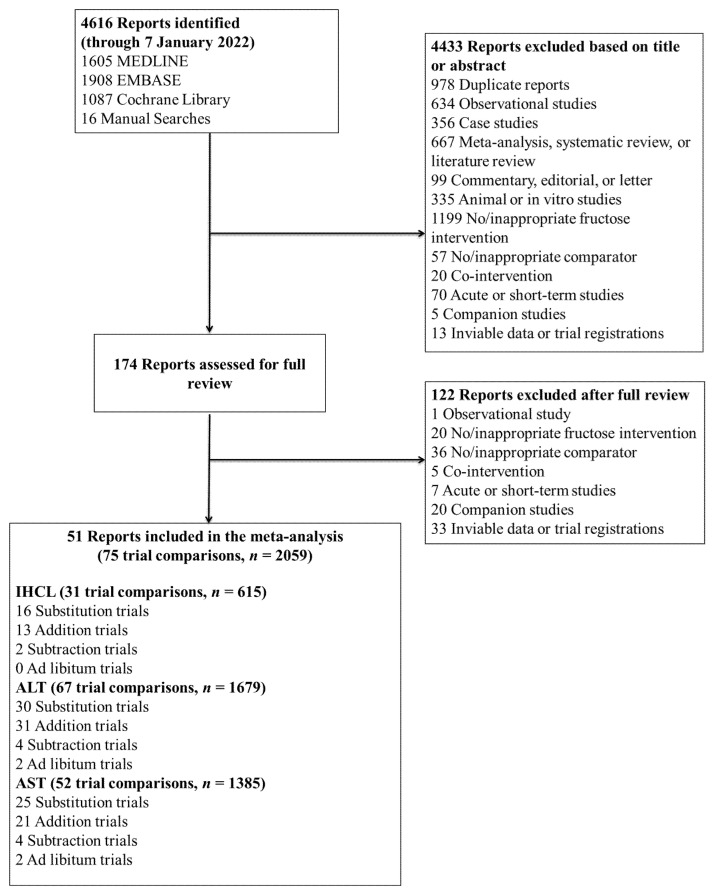
Flow of the literature for the effect of different food sources of fructose-containing sugars and NAFLD risk markers. ALT = alanine aminotransferase; AST = aspartate aminotransferase; IHCL = intrahepatocellular lipid.

**Figure 2 nutrients-14-02846-f002:**
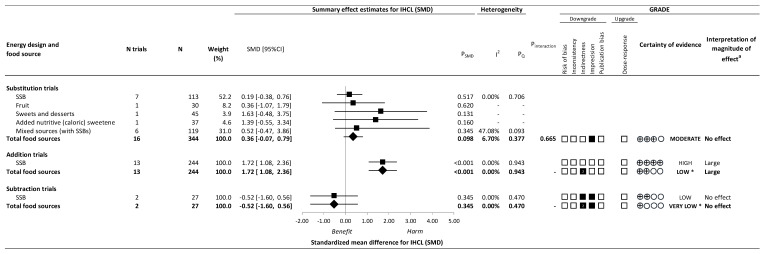
Summary plot for the effect of important food sources of fructose-containing sugars on intrahepatocellular lipid (IHCL). Data are weighted standardized mean differences (95% confidence intervals) for summary effects of individual food sources and total food sources on IHCL. Analyses conducted by generic, inverse variance random effects models (at least five trials available) or fixed effects models (fewer than five trials available). Between-study heterogeneity was assessed by the Cochrane Q statistic, where P_Q_ < 0.100 is considered statistically significant, and quantified by the *I*^2^ statistic, where *I*^2^ ≥ 50% is considered evidence of substantial heterogeneity. The Grading of Recommendations, Assessment, Development and Evaluation (GRADE) of randomized controlled trials are rated as “High” certainty of evidence and can be downgraded by five domains and upgraded by one domain. The white squares represent no downgrades, while filled black squares indicate a single downgrade or upgrades for each outcome, and the black square with a white “2” indicates a double downgrade for each outcome. CI = confidence interval; GRADE = Grading of Recommendations, Assessment, Development and Evaluation; IHCL = intrahepatocellular lipid; N = number; SMD = standardized mean difference; SSB = sugar-sweetened beverage. ^a^ For the interpretation of the magnitude, we used the MIDs to assess the importance of magnitude of our point estimate using the effect size categories according to new GRADE guidance. * Where there was a significant interaction by food source in substitution and addition trials and SSBs and mixed sources were the sole food sources in subtraction and ad libitum trials, we performed the GRADE analysis for each individual food source. To convert SMD to % liver fat, multiply the SMD by the baseline pooled standard deviation, 0.71%.

**Figure 3 nutrients-14-02846-f003:**
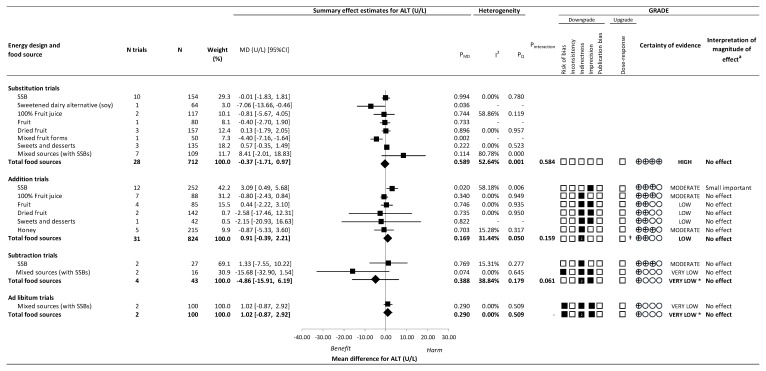
Summary plot for the effect of important food sources of fructose-containing sugars on alanine aminotransferase (ALT). Data are weighted mean differences (95% confidence intervals) for summary effects of individual food sources and total food sources on ALT. Analyses conducted by generic, inverse variance random effects models (at least five trials available) or fixed effects models (fewer than five trials available). Between-study heterogeneity was assessed by the Cochrane Q statistic, where P_Q_ < 0.100 is considered statistically significant, and quantified by the *I*^2^ statistic, where *I*^2^ ≥ 50% is considered evidence of substantial heterogeneity. The Grading of Recommendations, Assessment, Development and Evaluation (GRADE) of randomized controlled trials are rated as “High” certainty of evidence and can be downgraded by five domains and upgraded by one domain. The white squares represent no downgrades, while filled black squares indicate a single downgrade or upgrades for each outcome, and the black square with a white “2” indicates a double downgrade for each outcome. ALT = alanine aminotransferase; CI = confidence interval; GRADE = Grading of Recommendations, Assessment, Development and Evaluation; MD = mean difference; N = number; SMD = standardized mean difference; SSB = sugar-sweetened beverage. ^a^ For the interpretation of the magnitude, we used the MIDs to assess the importance of magnitude of our point estimate using the effect size categories according to new GRADE guidance. † Not upgraded for dose–response (see Appendix A for details). * Where there was a significant interaction by food source in substitution and addition trials and SSBs and mixed sources were the sole food sources in subtraction and ad libitum trials, we performed the GRADE analysis for each individual food source. To convert U/L to ukat/L, multiply U/L by 0.0167.

**Figure 4 nutrients-14-02846-f004:**
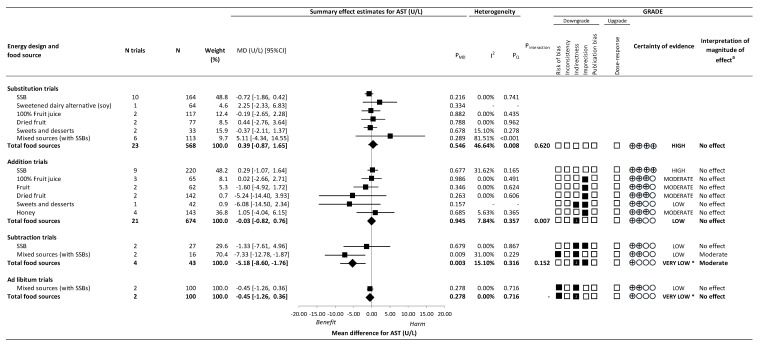
Summary plot for the effect of important food sources of fructose-containing sugars on aspartate aminotransferase (AST). Data are weighted mean differences (95% confidence intervals) for summary effects of individual food sources and total food sources on ALT. Analyses conducted by generic, inverse variance random effects models (at least five trials available) or fixed effects models (fewer than five trials available). Between-study heterogeneity was assessed by the Cochrane Q statistic, where P_Q_ < 0.100 is considered statistically significant, and quantified by the *I*^2^ statistic, where *I*^2^ ≥ 50% is considered evidence of substantial heterogeneity. The Grading of Recommendations, Assessment, Development and Evaluation (GRADE) of randomized controlled trials are rated as “High” certainty of evidence and can be downgraded by five domains and upgraded by one domain. The white squares represent no downgrades, while filled black squares indicate a single downgrade or upgrades for each outcome, and the black square with a white “2” indicates a double downgrade for each outcome. AST = aspartate aminotransferase; CI = confidence interval; GRADE = Grading of Recommendations, Assessment, Development and Evaluation; MD = mean difference; N = number; SMD = standardized mean difference; SSB = sugar-sweetened beverage. ^a^ For the interpretation of the magnitude, we used the MIDs to assess the importance of magnitude of our point estimate using the effect size categories according to new GRADE guidance. * Where there was a significant interaction by food source in substitution and addition trials and SSBs and mixed sources were the sole food sources in subtraction and ad libitum trials, we performed the GRADE analysis for each individual food source. To convert U/L to ukat/L, multiply U/L by 0.0167.

**Table 1 nutrients-14-02846-t001:** Summary of trial characteristics ^a^.

Trial Characteristics	Substitution Trials	Addition Trials	Subtraction Trials	Ad Libitum Trials
Trial comparisons (*n*)	35	39	4	2
Participants (median *n* ((range))	29 (7–102)	23 (6–91)	12 (5–15)	92 (92–92)
Health status (*n* trials)	NW = 5; MW = 7; OW/OB = 17; MetS = 1; NAFLD = 5	NW = 9; MW = 9; OW/OB = 4; DM2 = 3; NAFLD = 1; Other = 5	MW = 1; OW/OB = 3	MW = 2
Sex ratio (% male:female) ^b^	62:38	61:39	25:75	35:65
Age (years; median (range)) ^b^	38 (7.7–59)	36.2 (21.7–58.9)	28.7 (28.3–29.1)	29 (29–29)
Age category ratio (%; adult:children:mixed) ^b^	86:14:0	100:0:0	100:0:0	100:0:0
Country (*n* trials)	1 Brazil; 1 Denmark; 5 Finland; 2 Germany; 1 Greece; 2 Iran; 1 Sweden; 4 Switzerland; 7 UK; 8 USA; 1 Netherlands	1 China; 8 Denmark; 1 Germany; 2 Iran; 4 Malaysia; 2 Netherlands; 1 Pakistan; 1 Sweden; 8 Switzerland; 1 UK; 2 USA	2 Switzerland; 2 USA	2 Finland
Setting ratio (%; inpatients:outpatients:inpatients/outpatients)	9:88:3	8:87:5	50:50:0	0:100:0
Baseline IHCL (% liver fat; median (range)) ^c^	5.7 (2.6–16.5)	4.1 (1.6–9.7)	2.9 (2.6–3.3)	NR
Baseline ALT (U/L; median (range)) ^d^	23 (5.2–116.6)	25 (16.4–46.9)	28.5 (21.6–39.7)	NR
Baseline AST (U/L; median (range)) ^e^	25 (7.9–68.8)	26 (18–37)	24.7 (21.6–27.9)	NR
Trial design ratio (%; crossover:parallel)	37:62	56:44	50:50	0:100
Feeding control ratio (%; met:sup:DA:met/sup)	22:66:11:3	5:87:0:8	50:50:0:0	0:100:0:0
Randomization ratio (%; yes:no:partial) ^f^	91:9:0	85:15:0	50:50:0	0:0:100
Fructose-containing sugar dose (% of total energy intake; median (range))	20 (1–42)	12.2 (1.2–35)	16.3 (15–22.5)	14.4 (14–14.7)
Follow-up duration (median *n* (range) of weeks)	4 (1–52)	1 (1–24)	7 (1.7–12)	88 (88–88)
Funding sources (%; A:I:A+I:NR)	54:40:0:6	69:3:26:3	50:0:50:0	0:0:0:100
Fructose-containing sugar type (*n* trials)	Fructose = 11; fruit = 8; HFCS = 2; sucrose = 14	Fructose = 13; sucrose = 7; honey = 5; HFCS = 1; fruit = 13	Sucrose = 4	Fructose = 1; sucrose = 1
Sugar regulatory designation (*n* trials)	Naturally occurring = 7; added = 18; mixed = 10	Naturally occurring = 13; added = 26	Added = 2; mixed = 2	Mixed = 2
Comparator (*n* trials)	NNS = 1; fat = 8; glucose = 12; lactose = 2; starch = 5; mixed comparator = 9	NNS = 6; diet alone = 22; fat = 1; water = 3; other = 7	NNS = 4	NNS = 2
Food sources of fructose-containing sugars (*n* trials)	SSB = 13; sweetened dairy alternative (soy) = 1; 100% fruit juice = 2; fruit = 2; dried fruit = 3; honey = 1; sweets and desserts = 3; mixed sources (with SSBs) = 10	SSB = 20; 100% fruit juice = 7; fruit = 4; dried fruit = 2; honey = 5; sweets and desserts = 1	SSB = 2; mixed sources (with SSBs) = 2	Mixed sources (with SSBs) = 2

A = agency; A+I = agency and industry; ALT = alanine aminotransferase; AST = aspartate aminotransferase; Pre-CVD = pre-cardiovascular disease; DA = dietary advice; DM2 = type-2 diabetes mellitus; HIV = human immunodeficiency virus; I = industry; IHCL = intrahepatocellular lipid; met = metabolically controlled; MetS = metabolic syndrome; MW = mixed weight; NAFLD = non-alcoholic fatty liver disease; No = number; NR = not reported; NW = normal weight; OB = obese; ODM2 = overweight or obese type-2 diabetes mellitus; OW = overweight; SSBs = sugar-sweetened beverages; sup = supplemented; UK = United Kingdom; USA = United States of America. ^a^ Values are rounded to nearest whole number except for baseline NAFLD outcomes. ^b^ Based on trials which report data. ^c^ Based on trial comparisons that reported baseline data (*n* = 21 trials missing baseline IHCL substitution trials and *n* = 20 trials missing baseline IHCL addition trials). Baseline % liver fat was calculated by multiplying the baseline standardized mean difference of IHCL and baseline standard deviation of all trial comparisons that reported % liver fat. ^d^ Based on trial comparisons that reported baseline data (*n* = 9 trials missing baseline ALT substitution trials; *n* = 14 trials missing for baseline ALT addition trials; and *n* = 2 trials missing for baseline ALT ad libitum trials). ^e^ Based on trial comparisons that reported baseline data (*n* = 13 trials missing baseline AST substitution trials; *n* = 19 trials missing baseline AST addition trials; and *n* = 2 trials missing for baseline ALT ad libitum trials). ^f^ Partial randomization was assigned to a trial comparison which randomized only selected participants.

## Data Availability

Data available upon request.

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
