# Peer review of "Important Food Sources of Fructose-Containing Sugars and Non-Alcoholic Fatty Liver Disease: A Systematic Review and Meta-Analysis of Controlled Trials"

_nutrients, 2022, doi:10.3390/nu14142846_

Round 1

Reviewer 1 Report

The present systematic review and meta-analysis have a very good scientific soundness, good quality of presentation, and high complexity. 

The supplementary material is detailed and has a lot of information. In the article, the authors mention that details about the mechanisms underlying the potential effects of fructose-containing sugars and NAFLD markers are presented in Supplementary Table S9. However, some ideas about these mechanisms could be described also in the manuscript.

Author Response

Thank you for raising this important concern. We have added a short description to highlight the potential mechanisms within the manuscript. Please see lines 552 to 556 (4.2. Potential Mechanisms section).

Reviewer 2 Report

In this present study, the authors have conducted a Systematic Review and  Meta-Analysis of Controlled Trials with regard to the Important Food Sources of Fructose-Containing Sugars and Non-Alcoholic Fatty Liver Disease (NAFLD). This sounds interesting, however authors are requested to address the following queries:

1. The conceptual framework which is critical for this study is completely missing in this manuscript.

2. The introduction needs to be succinct and authors need to elaborate the introduction so as top set up the stage for the readers.

3. Authors have mentioned that the searches were supplemented with manual searches. Could the authors elaborate on this and provide more information on this?

4. Why only IHCL and ALT/AST were used as markers for NAFLD? Why not other specific markers like FIB-4 or APRI?

Author Response

Thank you for your review of our manuscript. We have responded to your comments below. We have also reviewed the manuscript for spelling and grammar.

1, 2. Thank you for highlighting the need for elaborating on the conceptual framework. We have modified the introduction accordingly. Please see lines 69-89.

  1. Thank you for commenting on the need to clarify how manual searches were performed. We have added text to explain our methodology. Please see lines 108-109 (2.1. Data Sources and Search Strategy section).
  2. The focus was on intermediate outcomes used in routine clinical assessments. Clinical practice guidelines  recommend assessing NAFLD by identifying hepatic steatosis (defined as ≥5% liver fat) using imaging techniques (where IHCL by proton MRS is the standard) or liver biopsies, in addition to routine blood panels of liver function tests, including ALT and AST, as part of regular check-ups (Canadian Liver Foundation , Chalasani et al., Heptatology 2018). Therefore, IHCL was selected as the primary outcome, and ALT and AST as secondary outcomes. We did not include FIB-4 and APRI as they are used less commonly in clinical assessments and are specific to more advanced (fibrosis) stages of NAFLD (Chalasani et al. 2018, Anstee et al., JHEP Reports 2022, Sebastiani et al., Can J Gastroenterol Hepatol 2014, Lee et al., Liver Int 2021). To this point, none of the identified studies in our search, including the 7 specifically in patients with NAFLD, reported FIB-4 or APRI.

Reviewer 3 Report

The authors submitted a comprehensive review and meta-analysis paper, which deals with relations between non-alcoholic fatty live disease and consummations of fructose rich food. Since live diseases are serious problem in population of various countries, the topic of this manuscript is important and interesting.

Strengths of the paper: The authors used systematic methods for data searching. Additionally, they selected the most relevant articles. Additionally, the authors adequately analysed the data and results, which are properly discussed.

Weaknesses: Since the review is comprehensive, it is easy for readers to become lost in many supplementary data. However, this is no serious weakness due to huge amount of very important data presented.

Author Response

Thank you for your comments. The supplementary file is indeed lengthy, but we agree that the data is important as to show transparency in our work.